# Potential of a Bead-Based Multiplex Assay for SARS-CoV-2 Antibody Detection

**DOI:** 10.3390/biology13040273

**Published:** 2024-04-18

**Authors:** Karla Rottmayer, Mandy Schwarze, Christian Jassoy, Ralf Hoffmann, Henry Loeffler-Wirth, Claudia Lehmann

**Affiliations:** 1Laboratory for Transplantation Immunology, University Hospital Leipzig, Universität Leipzig, Johannisallee 32, 04103 Leipzig, Germany; 2Institute of Bioanalytical Chemistry, Faculty of Chemistry and Mineralogy, Universität Leipzig, 04103 Leipzig, Germany; mandy.schwarze@uni-leipzig.de (M.S.);; 3Center for Biotechnology and Biomedicine, Universität Leipzig, 04103 Leipzig, Germany; 4Institute for Medical Microbiology and Virology, Leipzig University Hospital and Medical Faculty, University of Leipzig, Johannisallee 30, 04103 Leipzig, Germany; christian.jassoy@medizin.uni-leipzig.de; 5Interdisciplinary Centre for Bioinformatics, IZBI, Leipzig University, Haertelstr. 16-18, 04107 Leipzig, Germany; wirth@izbi.uni-leipzig.de

**Keywords:** SARS-CoV-2 immune response, validation, multiplex bead-based immunoassay, ELISA, neutralising antibodies

## Abstract

**Simple Summary:**

In this study, a bead-based multiplex assay for the detection of SARS-CoV-2 antibodies was validated in three study arms. Reproducibility was tested on *n* = 82 samples. In another arm of *n* = 30 samples, the assay was compared with several other SARS-CoV-2 antibody assays. In addition, the proportion of neutralising antibodies was determined in *n* = 58 samples. The bead-based multiplex assay is comparable to commercial ELISA/CLIA tests in terms of antibody detection and can be used to simultaneously detect antibodies against five SARS-CoV-2 domains and six other common cold coronaviruses. The bead-based test is sensitive to repeated freeze-thaw cycles and showed a decrease in reactions. With regard to the neutralising activity of RBD antibodies, we can show that the bead-based multiplex test provides the same results as the surrogate test.

**Abstract:**

Serological assays for SARS-CoV-2 play a pivotal role in the definition of whether patients are infected, the understanding of viral epidemiology, the screening of convalescent sera for therapeutic and prophylactic purposes, and in obtaining a better understanding of the immune response towards the virus. The aim of this study was to investigate the performance of a bead-based multiplex assay. This assay allowed for the simultaneous testing of IgG antibodies against SARS-CoV-2 spike, S1, S2, RBD, and nucleocapsid moieties and S1 of seasonal coronaviruses hCoV-22E, hCoV-HKU1, hCoV-NL63, and hCoV-OC43, as well as MERS and SARS-CoV. We compared the bead-based multiplex assay with commercial ELISA tests. We tested the sera of 27 SARS-CoV-2 PCR-positive individuals who were previously tested with different ELISA assays. Additionally, we investigated the reproducibility of the results by means of multiple testing of the same sera. Finally, the results were correlated with neutralising assays. In summary, the concordance of the qualitative results ranged between 78% and 96% depending on the ELISA assay and the specific antigen. Repeated freezing–thawing cycles resulted in reduced mean fluorescence intensity, while the storage period had no influence in this respect. In our test cohort, we detected up to 36% of sera positive for the development of neutralising antibodies, which is in concordance with the bead-based multiplex and IgG ELISA.

## 1. Introduction

At the beginning of the COVID-19 pandemic, the number and the development of various diagnostic assays dramatically increased. On the one hand, there was the need to clearly identify individuals infected with SARS-CoV-2 with optimal virus diagnostics and, on the other hand, the question of the individual immune response. The development of coronavirus vaccines primarily aimed to achieve basic immunity in the population and, according to the Standing Committee on Vaccination of Germany (STIKO), this is achieved if there have been at least three contacts with components of the coronavirus (antigens) [1]. Routine serodiagnosis and effective vaccination are essential for the development of sustainable immunity to SARS-CoV-2, including herd immunity. It is therefore essential to develop correspondingly specific and sensitive diagnostic test systems. In this context, serological tests for SARS-CoV-2 play a crucial role in understanding virus epidemiology and screening convalescent sera for therapeutic and prophylactic purposes. In addition, it is therefore possible to monitor the degree of sensitisation at an individual or population level. This opens up the assessment of the extent and duration of the immune response and, if necessary, to intervene in time through vaccination to protect the population.

To investigate the immune response against SARS-CoV-2, various assays are available that detect antibodies against SARS-CoV-2 [1,2], such as the enzyme-linked immunosorbent assay (ELISA) [3,4,5,6,7,8,9], chemiluminescence immunoassays (CLIAs) [3,7,10,11], or multiplex antibody detection assays [12]. The performance varies regarding the platform, sensitivity, specificity, and target antigens [2,13,14]. The consequence is that the comparability and interpretation of the results of different assays is not directly possible.

Neutralising antibodies (Nabs) are crucial for the inhibition of viral infection [8,15,16,17]. The formation of neutralising antibodies varies from individual to individual, as shown, for example, in the study by Pan et al. [18], where 78% of the specimens tested positive for neutralising antibodies, or the study by Liu et al. [19], which shows that about 17% of individuals did not produce NAbs after SARS-CoV-2 infection. In this regard, we evaluated the bead-based multiplex assay to determine whether the specific IgG antibodies detected correspond to NAbs. 

This study focused on the evaluation and validation of the following points: reproducibility of the results with long storage periods, concordance of the qualitative determination of positive and negative serum samples in comparison to commercial ELISA tests, and the correlation with NAbs titers according to Schwarze et al. [20].

## 2. Materials and Methods

### 2.1. Study Cohort

This test evaluation study was conducted as a part of a larger study [21,22]. For validation purposes, we analysed 82 samples from 52 subjects and examined the reproducibility of the results. The same sera have been retested (see Figure 1a). 

In addition, 30 samples of 30 individuals that had previously been tested for SARS-CoV-2 antibodies using various ELISA tests were analysed [6,9]. In total, 3 of these samples were blinded negative samples and 27 samples were SARS-CoV-2-infected individuals with known positive ELISA antibody test results by the provider. All test sera were provided by the Institute for Medical Microbiology and Virology, University Hospital and Medical Faculty University of Leipzig, and were once frozen at −20 °C [9].

A third sample comprising 58 sera from different subjects in the larger study was also tested for neutralising antibodies. Of these, 1 subject was vaccinated once, 2 subjects were vaccinated twice, 7 subjects were vaccinated three times, and 2 subjects were vaccinated four times against SARS-CoV-2. The assays were performed at the Institute of Bioanalytical Chemistry, Faculty of Chemistry and Mineralogy, Universität Leipzig, Germany Center for Biotechnology and Biomedicine.

The study was approved by the Local Ethics Commission at the Medical Faculty at the University of Leipzig (ethical vote 195/20-ek).

### 2.2. Serum Storage

All sera were centrifuged upon arrival at the laboratory, aliquoted at 500 µL, and frozen at −20 °C. Before testing, the sera were slowly thawed at room temperature and further tested according to the manufacturer’s instructions. From this thawed aliquot, 20 µL of serum was taken, and the aliquot was immediately frozen again at −20 °C. To test reproducibility, the same serum aliquot was thawed again on different days as described (Figure 1b). 

### 2.3. Bead-Based Immunoassay

In this study, the bead-based immunoassay LABScreen^TM^ COVID PLUS (One Lambda, West Hills, CA, USA) was validated and compared with other commercial ELISA tests for the detection of antibodies against SARS-CoV-2 [22,23]. LABScreen^TM^ COVID PLUS is an in vitro diagnostic flow cytometric antibody detection assay used for the qualitative detection of IgG antibodies against SARS-CoV-2 in human serum or plasma. LABScreen^TM^ COVID PLUS is an assay used to identify individuals with an adaptive immune response to SARS-CoV-2, indicating a recent or previous infection. 

The mean fluorescence intensity (MFI) was measured using the Luminex200^TM^ instrument system according to the manufacturer’s recommendation (Luminex^TM^, Corp., Austin, TX, USA) [12]. In addition to five specific SARS-CoV-2 antigens, six endemic coronavirus antigens against the S1 domain and SARS-CoV and MERS-CoV were determined simultaneously with the test kit. A defined negative control serum and positive control serum were included in the test runs (One Lambda, West Hills, CA, USA). At least 24 sera were tested per test run so that at least one test pack could be used up per test run. For quality control purposes, the laboratory successfully participated in external proficiency tests (RV416, INSTAND e.V., Düsseldorf, Germany). For further information on the application and use of the test kit, see also Rottmayer et al. [22].

Two different batches of the test were available, which differ in terms of cut-off values for each bead, which were taken based on the manufacturer’s recommendation (see Table 1). For the qualitative analysis, these cut-off values were used according to the batches.

### 2.4. Data Analysis of Bead-Based Immunoassay

The reactivity of a sample is calculated from the adjusted mean fluorescence values recorded by the LABScan device for each bead using xPONENT v4.3 software. Evaluation and normalisation were performed with HLA-Fusion^TM^ v4.5 software (Luminex, Corp., Austin, TX, USA); for examples, see Appendix A. To calculate adjusted sample-specific fluorescence values (baseline value), the HLA-Fusion^TM^ software assigned to each bead for each sample was tested according to the following formula:

Baseline value = (sample-specific fluorescence value for bead number N − sample-specific fluorescence value for negative control bead) − (background fluorescence value of the NC serum for bead no. N-background fluorescence value of the NC serum for negative control bead).

The cut-offs of the individual beads calculated by the manufacturer are directly assigned to the “positive” and “negative” results in colour by HLA-Fusion^TM^ software (Appendix A and Table 2). “Positive” is assigned if one of the LABScreen COVID PLUS SARS-CoV-2 bead regions has a baseline value that is higher than the specified cut-off in the lot-specific worksheet. “Negative” is assigned if all the LABScreen COVID PLUS SARS-CoV-2 bead regions have a baseline value that is lower than the specified cut-off values in the batch-specific worksheet.

### 2.5. IgG ELISA and ACE-2 Assay for the Detection of Neutralising Antibodies

Both methods have been described and used in previous studies [20,24]. For the IgG ELISA and the ACE-2 assay, 75 ng and 37.5 ng of recombinant expressed RBD protein per well were coated onto microtiter plates (12xF8, PS, F-bottomGreiner Bio-One, Frickenhausen, Germany;) in 100 µL of PBS (ROTI^®^Stock with 0.1 mol/L NaCl; Carl Roth GmbH & Co. KG, Karlsruhe, Germany) overnight at 4 °C. The plates were washed three times with 300 µL of PBS-T (ROTI^®^Stock) and blocked for 1 h with SuperBlock solution (Thermo Fisher Scientific, Waltham, MA, USA) at room temperature (RT). The plates were washed as described above, and the serum samples which were diluted 100-fold for the IgG ELISA or 10-fold for the ACE-2 assay in 100 µL of assay diluent (Surmodics IVD, Inc., Eden Prairie, MN, USA) were added and incubated for 45 min at RT. For the ACE-2 assay, 100 ng of recombinant, biotinylated ACE-2 protein (Sino Biological, Eschborn, Germany) was added to each well in 100 µL of PBS (37 °C, 45 min). Both the IgG ELISA and the ACE-2 assay were washed, as described before, and anti-human IgG-HRP (25,000-fold diluted, Promega GmbH, Walldorf, Germany) and ExtrAvidin-Peroxidase (4000-fold diluted, Sigma-Aldrich Chemie GmbH, St. Louis, MO, USA), respectively, were added to 100 µL of Stabilzyme (RT, 30 min, Surmodics IVD, Inc.). After three washing steps with PBS-T, TMB substrate solution (RT, 100 µL, Seramun Diagnostika GmbH, Heidesee, Germany) was added. After 10 min, the reaction was stopped by the addition of H_2_SO_4_ (1 mol/L, 100 µL, Carl Roth GmbH & Co. KG, Karlsruhe, Germany), and the absorbance (OD) was recorded at 450 nm using a microplate reader (SpectraMax Paradigm, Molecular Devices, Munich, Germany).

### 2.6. Statistical Analysis

The study employed the Wilcoxon rank-sum test to compare the antibody levels following different thawing cycles. Linear regression was used to analyse the effect of storage time on the MFI value as a percentage. Adjusted R^2^ values were calculated using Spearman’s method. Statistical significance was tested using Fisher’s two-tailed exact test. All statistical analyses were conducted using R version 4.2.3.

## 3. Results

### 3.1. Reproducibility of Antibody Measurement Using the Luminex Test and Its Dependence on the Number of Freeze–Thaw Cycles

Our first objective was to assess the reproducibility of antibody levels using the bead-based multiplex method. Therefore, we tested 82 sera at different time points and analysed the impact of the freeze–thaw cycles (FTCs). Ten serum samples were obtained from individuals who had not experienced any immunisation events, such as infection and/or vaccination. 

A significant decrease in antibody levels was observed in all domains with increasing FTCs. The median for the full spike showed a significant drop to 93% (*p*-value < 0.001) after the second FTC and to 74% after the third FTC (Figure 2a; *p*-value < 0.01). A stronger decrease was observed for the S1 and S2 domains, with S1 decreasing to 89% (Figure 2b; *p*-value < 0.001) after the second FTC and to 51% (Figure 2b, *p*-value < 0.001) after the third FTC, and with S2 decreasing to 81% (Figure 2d, *p*-value < 0.001) after the second FTC and to 48% after the third FTC (*p*-value < 0.01). The MFI value for the receptor binding domain remained unchanged after the second FTC but decreased to 62% after the third FTC (Figure 2c, *p*-value < 0.001). There was a significant decrease of 61% (Figure 2e, *p*-value < 0.001) in the nucleocapsid protein after the second FTC, but no significant decrease was observed between the second and third FTCs (*p*-values > 0.05; shown in Figure 2e). In order to analyse the variance of the MFI values at the respective FCTs, we calculated the coefficient of variance CV = standard deviation/mean × 100. Figure 2a–e show that the CVs increase with the FCTs.

In the following step, we examined whether the decrease could be attributed to the thawing cycles or if it was a result of the storage duration. To achieve this, we established the correlation between the percentage MFI values after the initial freeze–thaw cycle and time. The MFI values of the first measurement were set to 100% for this purpose. The same procedure was repeated following the freeze–thaw cycles. No linear correlation was observed for any domain or time point. Figure 3 displays the results for the full spike. The results for the other domains are available in the Appendix A. Therefore, it can be concluded that the decrease is a result of freezing and thawing.

The qualitative analysis of the antibody measurement shows that most sera are consistent in all domains after the second FTC, at 93% for spike and S1, 88% for RBD, 83% for S2, and 91% for Nc (positive–positive or negative–negative in Table 3). Furthermore, we observed a change in their qualitative results after the second FTC in a range from 7% (S1 and Nc) to 17% (S2). A further freeze–thaw cycle shows a lower number of sera with the same qualitative result of 65% for S2, 75% for S1, 77.5% for spike, 80% for RBD, and 82.5% for Nc.

All 10 sera obtained from individuals without prior immunisation tested negative and remained negative throughout the FTCs.

### 3.2. Validation of Antibody Measurement Using Luminex with Various Established ELISA/CLIA Tests

We compared the antibody detection of the Luminex multiplex bead-based immunoassay with commercial ELISA tests. Four different SARS CoV-2 antibody assays were previously compared [6] and served as comparative tests in this study: two for RBD IgG, one for S1 IgG, and four for nucleocapsid protein IgG. For the present analysis, 27 serum samples were analysed from individuals who tested positive for SARS-CoV-2 via PCR. In addition, all sera were previously examined for SARS-CoV-2 antibodies using the abovementioned ELISA/CLIA tests [6]. In addition, three negative serum samples were tested with concordant results compared with the nucleocapsid protein ELISA tests.

Throughout all the antibody assays, we found concordance ranging from 78% to 96%. The S1 ELISA showed the highest concordance (96%) in terms of the expected positive and negative reactions compared to the bead-based multiplex test (*p*-value = 0.0001; Figure 4c), while one of the RBD antibody tests showed the lowest concordance at 77.8% (Figure 4b). The bead-based multiplex test detected more positive antibody reactions for RBD (7% to 22%) compared to the S1 antibody assays shown in Figure 4a (*p*-value = 1.00) and Figure 4b (*p*-value = 0.798; marked in orange). Regarding the nucleocapsid protein, the concordance of the expected positive reactions varied between 59% (Figure 4f,g, *p*-values = 0.0001) and 70% (Figure 4d, *p*-value = 0.0007 and Figure 4e, *p*-value = 0.0006) depending on the different assays, while the concordance of the expected negative reactions varied between 18% and 30% depending on the ELISA/CLIA test. 

Table 4 displays the Spearman correlations between the Luminex assay and selected commercial ELISA/CLIA tests. Strong correlations were found with the Roche (0.68) and Abbott (0.79) tests, and very strong correlations were found with the assays from Siemens (0.87), Mediagnost (0.88), Euroimmun (0.95), Virotech (0.85), and Novatec (0.90). 

### 3.3. Proportion of Neutralising Antibodies in the Bead-Based Multiplex Test

The neutralising activity of antibodies is crucial in the immune defence against viruses such as SARS-CoV-2. Therefore, this study compared the qualitative antibody determination of the bead-based multiplex test with an RBD surrogate neutralising assay developed by Schwarze et al. [20,24] and the RBD IgG ELISA developed by the authors mentioned to confirm whether the detected RBD antibodies show neutralising activity. For this purpose, we tested 58 sera for neutralising antibodies, which were previously tested for SARS-CoV-2 antibodies in the bead-based assay.

In comparison to the neutralising assay, both the RBD IgG tests and the bead-based and neutralising antibody assay performed equally well (Figure 5a,b, *p*-values > 0.05). We observed a concordance of both assays of 46% (marked in blues and yellow). About 52% of the sera tested positive for IgG antibodies, but not in the neutralising assay (Figure 5a,b). Only one serum tested positive for antibodies in the neutralising assay, but not in the IgG tests (Figure 5b marked in grey).

A comparison of both IgG tests shows an 86% (81% positive–positive and 5% negative–negative) concordant antibody detection rate (*p*-value < 0.05; Figure 5c marked in blue and yellow). However, four sera were evaluated as positive by one test but not the other (Figure 5c marked in orange and grey).

## 4. Discussion

The immunoassay LABScreen^TM^ COVID PLUS offers the advantages of a multiplex test system, allowing for the simultaneous identification of patient immune responses against five SARS-CoV-2 proteins. As reported by Cox et al. [25], the test showed a 100% specificity and a sensitivity of 90% for the qualitative determination of the results. Furthermore, this test system can be used to simultaneously detect immune responses against the S1 domain of six other common cold coronaviruses, namely HCoV-229E, HCoV-HKU1, HCoV-NL63, HCoV-OC43, MERS-CoV, and SARS-CoV [26]. 

The Luminex^TM^ bead-based assay has shown a decrease in antibody detection after multiple freeze–thaw cycles. These results suggest that repeated freezing and thawing is unfavourable, and that aliquoting is necessary. Protein aggregation/degradation during freeze–thawing can be caused by various factors, for example, cold denaturation, pH shifts, phase separation, concentration of solutes, or ice formation [27]. Horn et al. [28] demonstrated that the effect is dependent on the pH during the FCTs. On the other hand, several publications indicate stable IgG antibody levels using ELISA test kits [29], which have also been published for SARS-CoV-2 antibodies [30], as well as their neutralising activity [31]. The results indicate that the decrease is an individual effect and domain-specific. However, for most of our samples (over 80%), the qualitative results remained the same across all domains. For the long-term storage of the sera, especially in the context of a biobank, it is recommended to store the sera in small aliquots at −20 °C or colder.

In addition, the bead-based multiplex assay is comparable to commercial ELISA/CLIA tests in antibody detection. This is in accordance with previous publications [32,33]. In fact, the bead-based assay performed better in detecting the RBD antibody in our examination and shows up to 22% more positive results compared to the ELISA (Figure 4b). 

With regard to the neutralising activity of RBD antibodies, we can show that the bead-based multiplex test provides the same results as the surrogate assay published by Schwarze et al. [20]. In the cohort studied, 41.2% of positive sera contained neutralising antibodies (Figure 5b). This suggests that over 60% of individuals developed no neutralising antibodies. The literature reports that 80–90% of people develop neutralising antibodies [33,34,35]. Neutralising antibodies peak during convalescence and then remain relatively stable for up to 15 months [35]. Favresse et al. [36] show that the neutralising antibody titers are higher in vaccinated individuals compared to SARS-CoV-2-infected people. With our study, we can confirm that all individuals with at least two vaccinations (*n* = 12) show a proportion of NAbs of >91% in contrast to those infected with SARS-CoV-2 only. Here, the proportion of NAbs could be determined in the range of 30–74%. On the other hand, Montesinos et al. reported a disappearance of NAbs in 16.9% of their participants within 6 months [34], and Tea et al. showed a decrease of 56% in the sera in a mildly affected cohort within 5 months [37].

A possible explanation for our rather low level of NAbs could be a dependence on disease severity, with more NAbs being formed in more severe disease, as reported by Petersen et al. [35]. However, our cohorts included moderately to mildly affected individuals. In the validation cohort, this value was confirmed using the bead-based multiplex assay (Figure 4a). A limitation of the bead-based assay is that the detection of RBD IgG antibodies does not allow for any statement about the individual proportion of neutralising antibodies to be made without an additional neutralising test. The bead-based multiplex assay can therefore be used primarily to diagnose general antibody production, i.e., to determine whether an infection has occurred. It can also be used to determine that the IgG is the main source of neutralising antibodies, as only one sample is neutralising, although it does not contain IgG. The neutralising activity could be assumed by IgA antibodies, which are specifically produced in the early immune response [38,39,40]. 

As the study by Cox et al. [25] shows, the multiplex technique has advantages such as effectiveness, it is time-saving, and has potential for immunity studies. The latter advantages were reported by Rottmayer et al. [22], in which a distinction was made between natural infection and vaccine response, and individual patterns in the immune response were described. Despite these advantages, there are limitations to the bead-based method. The MFI values are relative values [41], which means that a quantitative and direct comparison with other methods is not possible. In order to be able to compare the results with other test systems not only qualitatively, all the test systems should be normalised to one and the same measure (e.g., international units).

Furthermore, it is important to note that our study is limited by the lack of pre-COVID-19 pandemic samples. Therefore, we were only able to test three confirmed negative samples provided by the virology department, all of which were also negative when confirmed by the multiplex bead-based assay. The assay used here was CE-labelled. In the development paper by Bray et al. [12], they showed a specificity and sensitivity of 98.6% and 100%, respectively, in SARS-CoV-2-positive and -negative samples, including pre-COVID-19 pandemic samples.

The individual immune response and antibody formation are influenced by the severity of the disease and pre-existing conditions, such as obesity, diabetes, and chronic kidney disease [42]. Previous medical history was not documented. Therefore, the impact of pre-existing conditions on the variance of antibody measurements cannot be reported and is regarded as a limitation of this study.

Antibody testing, even with the multiplex bead-based assay, provides a snapshot and is subject to the influence of antibody kinetics. Therefore, negative antibody results may be due to the antibodies not yet being detectable or no longer being detectable. The antibody tests in our study were conducted, on average, 70 days after the PCR test [22]. This timeframe falls between the seroconversion period of one to three weeks, as described in the literature [43], and a phase with stable antibody levels, which can last for up to 9 months [44].

## 5. Conclusions

In conclusion, the bead-based assay is a fast assay for the qualitative determination of several antigenic determinants of SARS-CoV-2, including the simultaneous detection of common cold coronavirus antibodies. This allows individual patterns in the immune response to be recognised and cross-reactions to be determined, as shown in [22].

The RT-PCR test is considered to be the most reliable method to clarify the suspicion of an acute infection with the SARS-CoV-2 coronavirus. However, there are limitations that can lead to false-negative and false-positive results due to timing or the inaccurate collection of swab samples [45,46]. Therefore, serological testing has been suggested as a complement assay to RT-PCR to compensate for the RT-PCR limitations. Serological monitoring of the patient’s antibody response can provide information on virus contact, the individual immune response, and its development [45,46,47]. By combining RT-PCR technology as a diagnostic tool and serological assays as long-term monitoring tools, the understanding of viral epidemiology can be researched on the one hand, and the picture of herd immunity in the population can be monitored on the other. This allows for better decisions to be made about vaccination strategies. 

## Figures and Tables

**Figure 1 biology-13-00273-f001:**
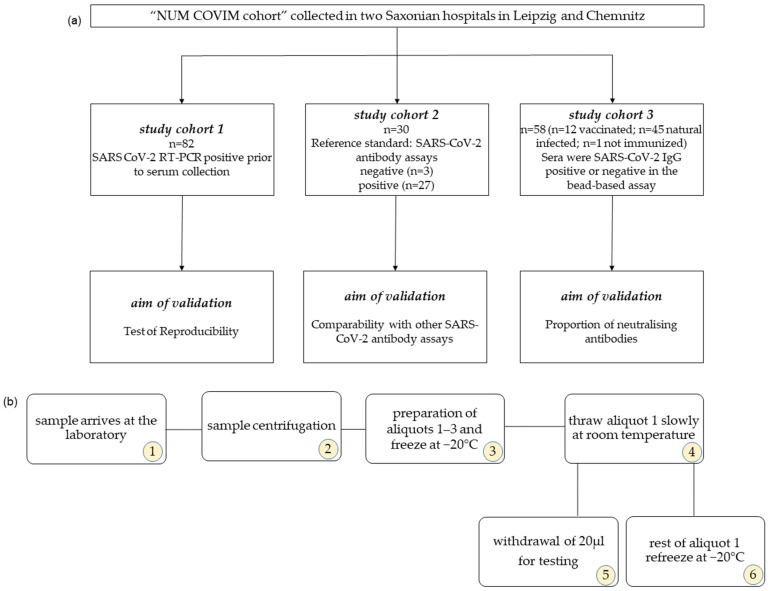
(**a**) Overview of the validation cohorts. (**b**) Diagram of the freezing and thawing cycles. The numbers in the circle indicate the sequence of the individual work steps. For multiple freezing and thawing cycles, steps 5 and 6 are repeated accordingly.

**Figure 2 biology-13-00273-f002:**
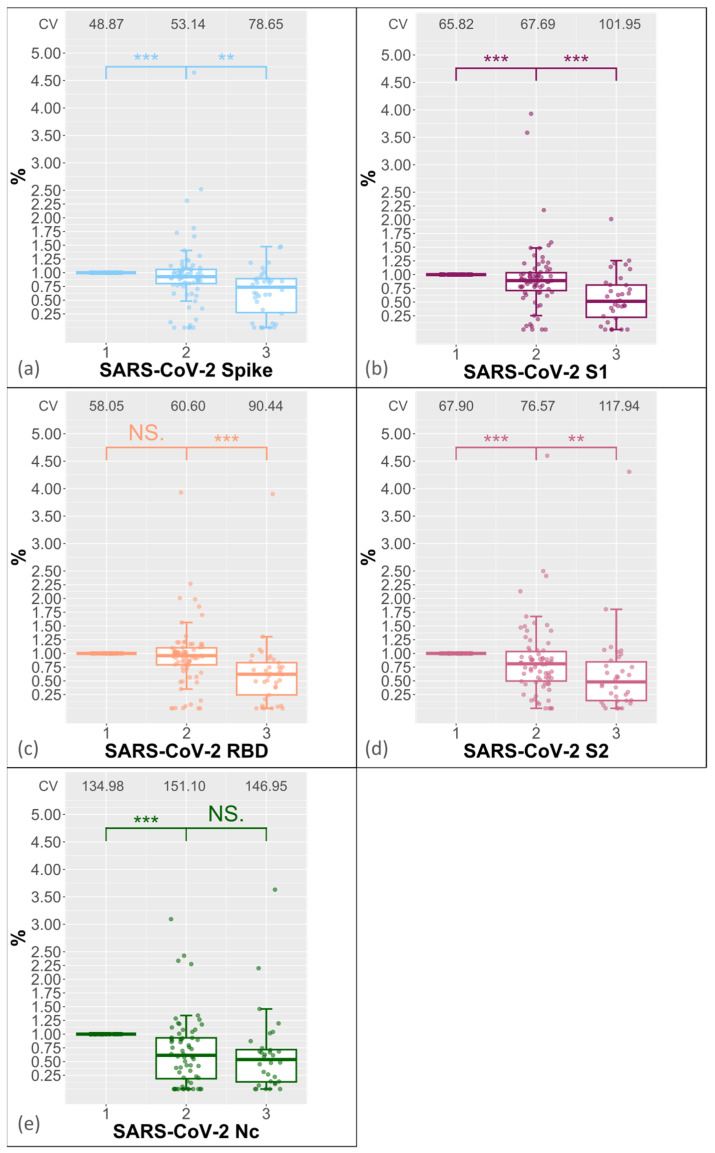
The boxplot displays the percentage decrease in antibodies against various domains of SARS-CoV-2, including the (**a**) full spike, (**b**) S1, (**c**) RBD, (**d**) S2, and (**e**) nucleocapsid protein, with each freeze–thaw cycle. Time 1 represents the percentage MFI value after first FTC, time 2 represents the percentage MFI value after the second FTC, and time 3 represents the percentage MFI value after the third FTC. *** indicate a *p*-value < 0.001, and ** indicate a *p*-value < 0.01. The coefficients of variance (CV) of the domain-specific MFI values are shown at the top for each FTC. NS indicates “not significant”.

**Figure 3 biology-13-00273-f003:**
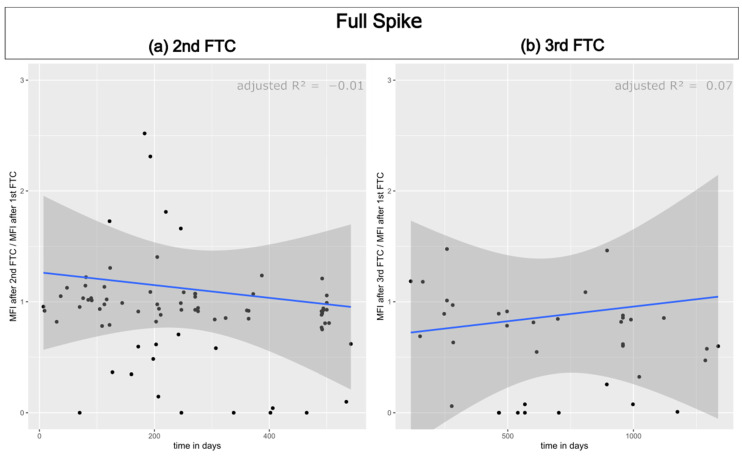
The correlation between the percentage MFI values of the full spike antibody and the storage duration was examined. The results are shown in (**a**) for the sera after the second freeze–thaw cycle and in (**b**) for the sera after the third freeze–thaw cycle. The linear regression equation and adjusted R^2^ values are displayed in the top right corner. The black dots mark individual serum samples.

**Figure 4 biology-13-00273-f004:**
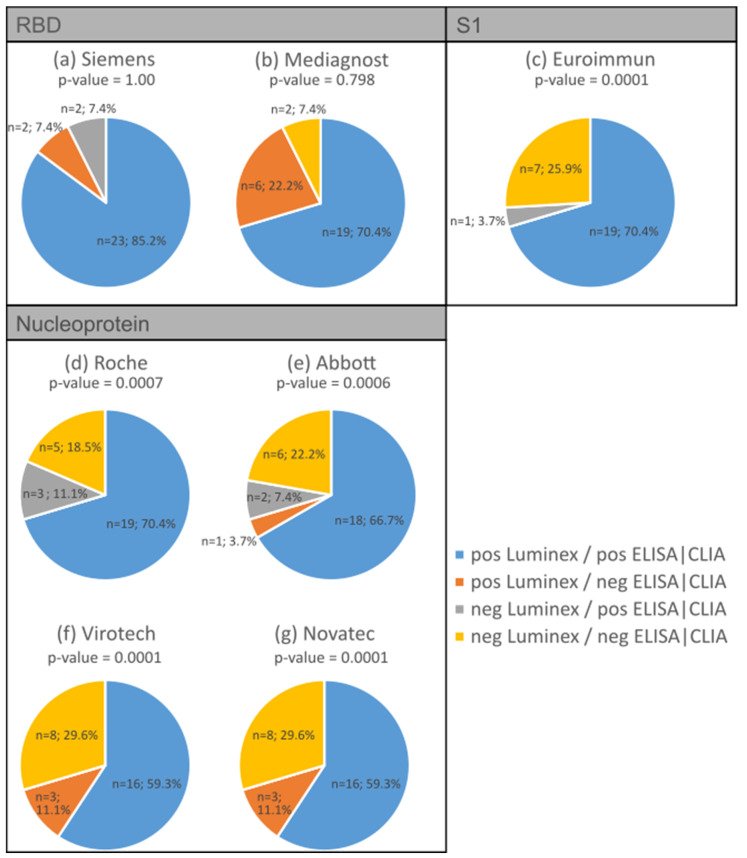
Comparison between the antibody detection using the bead-based multiplexing method and various commercial ELISA/CLIA tests: RBD IgG assay from Siemens (**a**) and Mediagnost (**b**); S1 IgG ELISA from Euroimmun (**c**) and nucleocapsid protein antibody assay from Roche (**d**), Abbott (**e**), Virotech (**f**), and Novatech (**g**). Concordance is represented by blue and yellow, while disconcordance is represented by orange and grey.

**Figure 5 biology-13-00273-f005:**
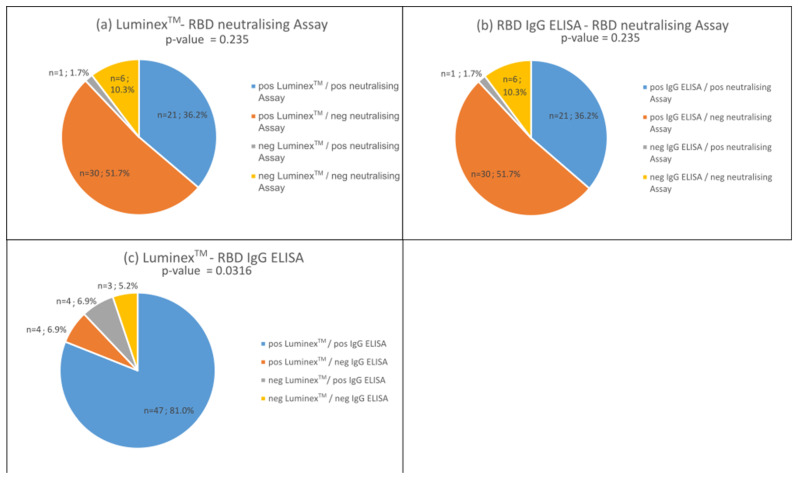
Comparison of sera (*n* = 58) for RBD IgG antibody detection using the bead-based multiplex assay (Luminex^TM^) and the ELISA developed by Schwarze et al. [1] with a neutralisation test developed by the same authors. (**a**) Bead-based multiplex assay compared to neutralising assay. (**b**) ELISA compared with neutralisation test. (**c**) Comparison of bead-based multiplex assay (Luminex^TM^) and ELISA tests. Concordance is represented by blue and yellow, while disconcordance is represented by orange and grey.

**Table 1 biology-13-00273-t001:** MFI cut-off values of LABScreen^TM^ COVID PLUS in batch 1 and 2 ^1^.

Antigen	MFI Cut-Off Batch 001	MFI Cut-Off Batch 002
SARS-CoV-2 Spike	7500	6800
SARS-CoV-2 Spike S1	4000	2700
SARS-CoV-2 Spike RBD	3500	5800
SARS-CoV-2 Spike S2	1900	3200
SARS-CoV-2 Nucleocapsid Protein	3500	5900
HCoV-229E Spike S1	3068	8012
HCoV-HKU1 Spike S1	2614	4235
HCoV-NL63 Spike S1	1043	4407
HCoV-OC43 Spike S1	3127	3599
MERS-CoV Spike S1	10	21
SARS-CoV Spike S1	92	41

^1^ not valid for LABScan3D^TM^.

**Table 2 biology-13-00273-t002:** LABScreen COVID PLUS reaction assignment.

Negative Control Bead (Bead-ID 1)	Positive Control Bead (Bead-ID 2)	SARS-CoV-2 Bead	Sample
−	+	+	positive
−	+	−	negative
−	−	+/−	invalid test (retested)
+	+	+/−	invalid test (retested)
+	−	+/−	invalid test (retested)

**Table 3 biology-13-00273-t003:** Influence of thawing cycles on the qualitative result of the specific SARS-CoV-2 antigens (sera numbers).

Spike	2nd Freeze–Thaw Cycle	3rd Freeze–Thaw Cycle
positive–positive	64	22
positive–negative	4	8
negative–positive	2	1
negative–negative	12	9
**S1**	**2nd freeze–thaw cycle**	**3rd freeze–thaw cycle**
positive–positive	62	21
positive–negative	5	9
negative–positive	1	1
negative–negative	14	9
**RBD**	**2nd freeze–thaw cycle**	**3rd freeze–thaw cycle**
positive–positive	59	20
positive–negative	5	7
negative–positive	3	1
negative–negative	13	12
**S2**	**2nd freeze–thaw cycle**	**3rd freeze–thaw cycle**
positive–positive	55	16
positive–negative	11	12
negative–positive	3	2
negative–negative	13	10
**Nc**	**2nd freeze–thaw cycle**	**3rd freeze–thaw cycle**
positive–positive	17	6
positive–negative	6	7
negative–positive	1	0
negative–negative	58	27

**Table 4 biology-13-00273-t004:** Spearman correlation between the Luminex assay and selected commercial ELISA tests.

SARS-CoV-2 Domain	ELISA/CLIA Test	Spearman’s Correlation Coefficient
RBD	Siemens *	0.87
	Mediagnost	0.88
S1	Euroimmun	0.95
Nucleoprotein	Roche *	0.68
	Abbott *	0.79
	Virotech	0.85
	Novatec	0.90

* indicate CLIA tests.

## Data Availability

The data supporting this study’s findings are available from the corresponding authors upon reasonable request.

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
