# Peer review of "Potential of a Bead-Based Multiplex Assay for SARS-CoV-2 Antibody Detection"

_biology, 2024, doi:10.3390/biology13040273_

Round 1
Reviewer 1 Report
Comments and Suggestions for Authors
The manuscript authored by Rottmayer and colleagues presents findings on bead-based multiplex assay for SARS-CoV-2 antibody detection.
Therefore, this manuscript by Rottmayer et al. has the major findings:
“1) The Reproducibility of antibody measurement using the bead-based multiplex assay for SARS-CoV-2 antibody detection test s dependence on the number of freeze-thaw cycles;
2) The bead-based immunoassay was validated through comparison with various established ELISA tests;
3) The authors determine the proportion of neutralizing antibodies in the bead based multiplex test.”
The authors' study is relevant in the context of SARS-CoV-2 diagnostic. However as highlighted below, overall, the methodology and Discussion would benefit from a more coherent structure and concise explanation of the researches. Thus, I have several points that I see interesting to add to this manuscript:
Major comments
Materials and Methods: The methodology part was not clear. I suggest adding a flowchart with all the steps performed to make it clearer to the reader.
Materials and Methods (lines 87-96): Since the main focus of the study is the bead-based immunoassay, it is important for the authors to clarify the protocol for this test.
Results: A Bayesian analysis and frequentist latent class models may be interesting to authors as it constitutes a valid alternative for estimating test accuracy in the absence of a gold standard.
Discussion: The authors must discuss the limitations of the bead-based multiplex assay in terms of low specificity and sensitivity when comparing the detection of neutralizing antibodies to SARS-CoV-2 (figure 4a).
Discussion: It is important for the authors to discuss that the gold standard in SARS-CoV-2 detection is RT-PCR, and that serological tests are prone to cross-reactivity.
Discussion: It is important for the authors to formulate a paragraph regarding the limitations of the study.
Minor comments
Introduction (line 49): Please considerer writing “SARS-CoV-2” instead of “SARS-CoV-2 virus”.
Materials and Methods (lines 97-99): It is important for the authors to clarify how the cut-off values were calculated.
Author Response
Thank you very much for your comments. We provided a point by point answer, attached.

Reviewer 2 Report
Comments and Suggestions for Authors
Serological assays for SARS-CoV-2 play a pivotal role in defining whether patients are infected, understanding viral epidemiology, screening convalescent sera for therapeutic and prophylactic purposes, and better understanding the immune response toward the virus. This study aimed at evaluating the performance of a bead-based multiplex assay that allowed simultaneous testing of IgG antibodies against SARS-CoV-2 spike, S1, S2, RBD, and nucleocapsid moieties and S1 of seasonal coronaviruses hCoV-22E, hCoV-HKU1, hCoV-NL63, and hCoV-20 OC43, as well as MERS and SARS-CoV. The authors compared the bead-based multiplex assay with commercial ELISA tests. They tested the sera of SARS-CoV-2 PCR-positive individuals, which were previously tested with different ELISA assays. In addition, they investigated the reproducibility of the results using multiple tests of the same sera and the correlation of the results with neutralizing assays. Unfortunately, there are major concerns in the design of the study. The comments and suggestions are listed below.
1. Lines 62-65: One of the major goals of this study is to evaluate the reproducibility of the results with long-term storage periods. However, each test's CV or SD value was missing. Without the data, it is impossible to evaluate the reproducibility of the test. In addition, to evaluate the impact of freeze-thaw cycles (FTCs) on the assays, it is critical to follow freeze-thawing SOPs because different ways to freeze and thaw a sample will have very different effects. However, the procedure was not described at all.
2. Line 100: Table 1 shows the MFI cut-off values of LABScreen COVD PLUS in batches 1 and 2. However, the acceptable criteria for each sample result and the criteria for determining whether samples were positive or negative were not described.
3. Lines 126-129: R2 values best indicate the correlation between two assay results. Providing R2 values to evaluate the correlation between the beads assay data and the ELISA data would be much more informative.
4. Vox et al. published a paper in the Journal of Immunological Methods (https://doi.org/10.1016/j.jim.2023.11347). The paper's title is “Comparative evaluation of Luminex-based assays for detecting SARS-CoV-2 antibodies in a transplantation laboratory.” This paper should be cited in this manuscript, and the authors should discuss what new information will be provided.
Author Response

(The authors gave the same response as above.)

Round 2
Reviewer 1 Report
Comments and Suggestions for Authors
The authors made changes that improved the manuscript.
Reviewer 2 Report
Comments and Suggestions for Authors
The authors answered all my comments and suggestions thoroughly. The quality of the revised manuscript is significantly improved.